 **eLIFE**

# A homozygous loss-of-function *CAMK2A* mutation causes growth delay, frequent seizures and severe intellectual disability

Poh Hui Chia[1†*], Franklin Lei Zhong[1,2†*], Shinsuke Niwa[3,4†], Carine Bonnard[1], Kagistia Hana Utami[5], Ruizhu Zeng[5], Hane Lee[6,7], Ascia Eskin[6,7], Stanley F Nelson[6,7], William H Xie[1], Samah Al-Tawalbeh[8], Mohammad El-Khateeb[9], Mohammad Shboul[10], Mahmoud A Pouladi[5,11], Mohammed Al-Raqad[8], Bruno Reversade[1,2,12,13*]

[1]Institute of Medical Biology, Immunos, Singapore; [2]Institute of Molecular and Cell Biology, Proteos, Singapore; [3]Frontier Research Institute for Interdisciplinary Sciences, Tohoku University, Sendai, Japan; [4]Graduate School of Life Sciences, Tohoku University, Sendai, Japan; [5]Translational Laboratory in Genetic Medicine, Agency for Science, Technology and Research, Singapore, Singapore; [6]Department of Pathology and Laboratory Medicine, David Geffen School of Medicine, University of California, Los Angeles, Los Angeles, United States; [7]Department of Human Genetics, David Geffen School of Medicine University of California, Los Angeles, Los Angeles, United States; [8]Queen Rania Paediatric Hospital, King Hussein Medical Centre, Royal Medical Services, Amman, Jordan; [9]National Center for Diabetes, Endocrinology and Genetics, Amman, Jordan; [10]Al-Balqa Applied University, Faculty of Science, Al-Salt, Jordan; [11]Department of Medicine, Yong Loo Lin School of Medicine, National University of Singapore, Singapore, Singapore; [12]Department of Paediatrics, National University of Singapore, Singapore, Singapore; [13]Medical Genetics Department, Koç University School of Medicine, Istanbul, Turkey

*For correspondence:
Pohhui.chia@reversade.com
(PHC);
franklin.zhong@reversade.com
(FLZ);
bruno@reversade.com (BR)

†These authors contributed equally to this work

Competing interests: The authors declare that no competing interests exist.

**Abstract** Calcium/calmodulin-dependent protein kinase II (CAMK2) plays fundamental roles in synaptic plasticity that underlies learning and memory. Here, we describe a new recessive neurodevelopmental syndrome with global developmental delay, seizures and intellectual disability. Using linkage analysis and exome sequencing, we found that this disease maps to chromosome 5q31.1-q34 and is caused by a biallelic germline mutation in *CAMK2A*. The missense mutation, p. His477Tyr is located in the CAMK2A association domain that is critical for its function and localization. Biochemically, the p.His477Tyr mutant is defective in self-oligomerization and unable to assemble into the multimeric holoenzyme. *In vivo*, CAMK2A$^{H477Y}$ failed to rescue neuronal defects in *C. elegans* lacking *unc-43*, the ortholog of human *CAMK2A*. *In vitro*, neurons derived from patient iPSCs displayed profound synaptic defects. Together, our data demonstrate that a recessive germline mutation in *CAMK2A* leads to neurodevelopmental defects in humans and suggest that dysfunctional CAMK2 paralogs may contribute to other neurological disorders.
DOI: https://doi.org/10.7554/eLife.32451.001

## Introduction

Calcium/calmodulin-dependent protein kinase II (CAMK2) is a calcium-activated serine/threonine kinase that is extremely abundant in the brain, comprising as much as 0.3% of the total brain protein content (*Bennett et al., 1983*). CAMK2 is highly enriched at the synapses and is necessary for the

**eLife digest** Each year, some children are born with developmental disorders and intellectual disabilities. These conditions are often caused by mutations in specific genes. Sometimes both copies of a gene – one inherited from each parent – need to be mutated for the symptoms to develop. These mutations are known as recessive mutations.

Here, Chia, Zhong, Niwa et al. diagnosed two siblings in their clinical care with a new form of neurological disease that affects the development of the brain and leads to frequent seizures. To test whether the young patients shared a genetic mutation that could explain their condition, the researchers analyzed the DNA of the children and compared the results with the DNA from their parents and healthy siblings. The results showed that the two children with the condition had inherited a recessive mutation in a gene called *CAMK2A*. The protein this gene encodes helps nerve cells to form connections and communicate with each other, and it has been shown to be essential for learning and memory.

The CAMK2A enzyme is made up of several identical subunits that form a complex. Chia et al. discovered that the mutation prevented these subunits from joining together properly, resulting in a faulty protein. *CAMK2A* and other related proteins are crucial for the health of the brain in a wide range of animals. Indeed, experiments in *Caenorhabditis elegans*, a roundworm commonly used to study neurons, confirmed that the mutation inherited by the children indeed caused similar neurological defects in the worms. Taken together, these experiments suggest that the children's condition is caused by the mutation in both copies of the *CAMK2A* gene.

For patients born with inherited diseases, it is often difficult to pinpoint exactly which mutation is responsible for the specific disorder. These findings could therefore help pediatric geneticists recognize this newly defined syndrome and reach the correct diagnoses. These results could also be the starting point for studies that look into restoring the activity of the defective CAMK2A protein. More broadly, identifying genes that are critical for the healthy development of the brain could shed light on common neurological conditions, such as epilepsy and autism.

DOI: https://doi.org/10.7554/eLife.32451.002

process of long-term potentiation (LTP), the activity-dependent strengthening and modulation of synaptic activity that is thought to be the molecular basis of some forms of learning and memory (*Kandel et al., 2014*; *Lisman et al., 2002*). In humans, there are four genes encoding distinct CAMK2 iso-enzymes. CAMK2A and CAMK2B are the predominant isoforms in the nervous system, with CAMK2A being expressed 3–4 times higher than CAMK2B (*Hanson and Schulman, 1992*). Each enzyme comprises a kinase domain, a regulatory domain and an association domain. Structurally CAMK2 holoenzymes are homo- or hetero-oligomers, consisting of 12 or 14 CAMK2 subunits (*Hudmon and Schulman, 2002b*). The holoenzyme assembly requires the carboxy-terminal association domain, which forms stacked pairs of hexameric or heptameric rings with the regulatory and kinase domain projecting radially to interact with other essential proteins for CAMK2 function and localization (*Bhattacharyya et al., 2016*). In the absence of calcium signalling, CAMK2 is inactive, as the regulatory domain inhibits kinase function (*Yang and Schulman, 1999*). This auto-inhibition is relieved when calcium-loaded calmodulin binds to the regulatory domain, thereby exposing the kinase domain and allowing it to phosphorylate target substrates (*Meador et al., 1993*). $Ca^{2+}$-Calmodulin binding also exposes Thr286 on the regulatory domain of CAMK2A, which becomes phosphorylated in trans by adjacent subunits. Once this residue is phosphorylated, the enzyme is persistently active, even in the absence of continual $Ca^{2+}$-Calmodulin signaling (*Rich and Schulman, 1998*; *Stratton et al., 2013*). This switch from auto-inhibition to autonomous, persistent activity is thought to constitute a biochemical form of memory, which marks the neuron for having experienced a previous calcium influx (*Bhattacharyya et al., 2016*; *Stratton et al., 2014*; *Stratton et al., 2013*).

Mice that are homozygous null for *CAMK2A* are viable and display impaired spatial memory and reduced LTP in the hippocampus (*Silva et al., 1992a*, *1992b*). The heterozygous mutant (*Camk2a$^{+/-}$*) show a significant deficit in spatial working memory and contextual fear memory (*Frankland et al., 2001*; *Matsuo et al., 2009*). More recently, it was demonstrated that mice with neuron-specific

conditional knock-out of *CAMK2A* similarly displayed learning deficits and defects in LTP that were comparable to the complete knockout mice (*Achterberg et al., 2014*). These findings suggest that neuron-intrinsic CAMK2A function is indispensable during the period of learning for memory formation. CAMK2 is conserved in invertebrates, such as *D. melanogaster* and *C. elegans*, where the kinase also plays critical roles in behavioral and cognitive traits (*Cho et al., 1991*; *Hudmon and Schulman, 2002a*; *Reiner et al., 1999*; *Rongo and Kaplan, 1999*). In *C. elegans*, the only CAMK2 is encoded by the *unc-43* gene, which is essential for synaptic function (*Rongo and Kaplan, 1999*). Loss of *unc-43* causes worms to have flaccid muscle tone, locomotion defects and spontaneous body contractions that resemble seizures (*Reiner et al., 1999*).

In pediatric care, global developmental delay in infants is defined as a significant functional delay in two or more developmental domains including gross and fine motor function, speech and language, cognition, social development and personal skills (*Quality Standards Subcommittee of the American Academy of Neurology et al., 2003*). These defects are detected at an early age in children age five years or under, and can persist throughout life (*Shevell, 2008*). About 25–50% of identified case are caused by germline genetic changes, including chromosomal abnormalities, copy number variants and monogenic mutations (*Srour and Shevell, 2014*; *van Bokhoven, 2011*). For many patients with global neuro-developmental delay, the genetic etiology remains unknown.

## Results

Here, we report the identification of a consanguineous family from Jordan with two affected children manifesting global neuro-developmental delay with frequent seizures and convulsions. The two affected siblings had no dysmorphic features but failed to develop the ability to walk or speak (*Figure 1A,B*, *Figure 1—figure supplement 1A*). They displayed progressive psychomotor retardation with hypotonic muscles (Supplemental Material, *Videos 1* and *2*). Electroencephalogram (EEG) analysis revealed abnormal epileptiform transients (*Figure 1C*, *Figure 1—figure supplement 1B*), consistent with frequent myoclonic seizures. Magnetic resonance imaging (MRI) scan showed no major structural defects in the brain of proband II:4 (*Figure 1—figure supplement 1C*). Serum metabolite levels were normal, ruling out potential lysosomal storage disorders.

Assuming autosomal recessive inheritance, we performed identity-by-descent (IBD) homozygosity mapping using genomic DNA from both parents, two affected probands and three healthy siblings. Only one candidate locus greater than 5 cM on chromosome 5 spanning 28 Mb was delineated (*Figure 1D*, *Figure 1—figure supplement 1D*). Whole-exome sequencing was subsequently performed on index case II:1. After filtering for variants with low quality and low sequencing coverage, 72 homozygous variants were identified, out of which four lie within the Chr. 5 IBD region (*Table 1*). Three of the homozygous variants had been previously annotated as known polymorphisms with minor allele frequencies > 0.0001. In addition, healthy individuals who are homozygous for the minor alleles had been identified in genomic sequencing databases such as dbSNP and gnomAD (*Figure 1—figure supplement 1E*). We therefore filtered out these variants as non-pathogenic. The fourth homozygous variant within the IBD region mapped to the *CAMK2A* gene (MIM: 114078), resulting in a missense mutation p.His477Tyr that has never been observed in previous large-scale sequencing databases and our in-house ethnically-matched cohort (*Figure 1E*, *Figure 1—figure supplement 1E*). Using Sanger sequencing, this private mutation was confirmed to segregate with the disease in all seven family members (*Figure 1A,D*).

CAMK2A is a neuron-specific, highly abundant serine/threonine kinase that plays critical roles in synaptic plasticity. To understand how neuronal function is affected due to the mutation in CAMK2A, we reprogrammed primary dermal fibroblasts from patient II.4 into iPSCs, differentiated them into neurons and measured population-level neuronal activity using a multi-electrode array (MEA) system (*Figure 2A*). Compared to H1 embryonic stem cell derived neurons and an unrelated CAMK2A wild-type iPSC control, the patient's iPSCs were equally efficient in differentiating into mature neurons expressing neuronal markers TUJ1 and MAP2 after 21 days of *in vitro* differentiation (*Figure 2B*). When these differentiated neurons were plated on MEA plates to measure spontaneous action potentials, we observed a significant reduction in both the total number of spontaneous spikes (*Figure 2C*, left) and the mean firing rate (*Figure 2C*, right) in the patient-derived neurons harboring the p.H477Y mutation compared to the wild-type controls, suggesting that CAMK2A[H477Y] causes a profound functional defect in cultured neurons.

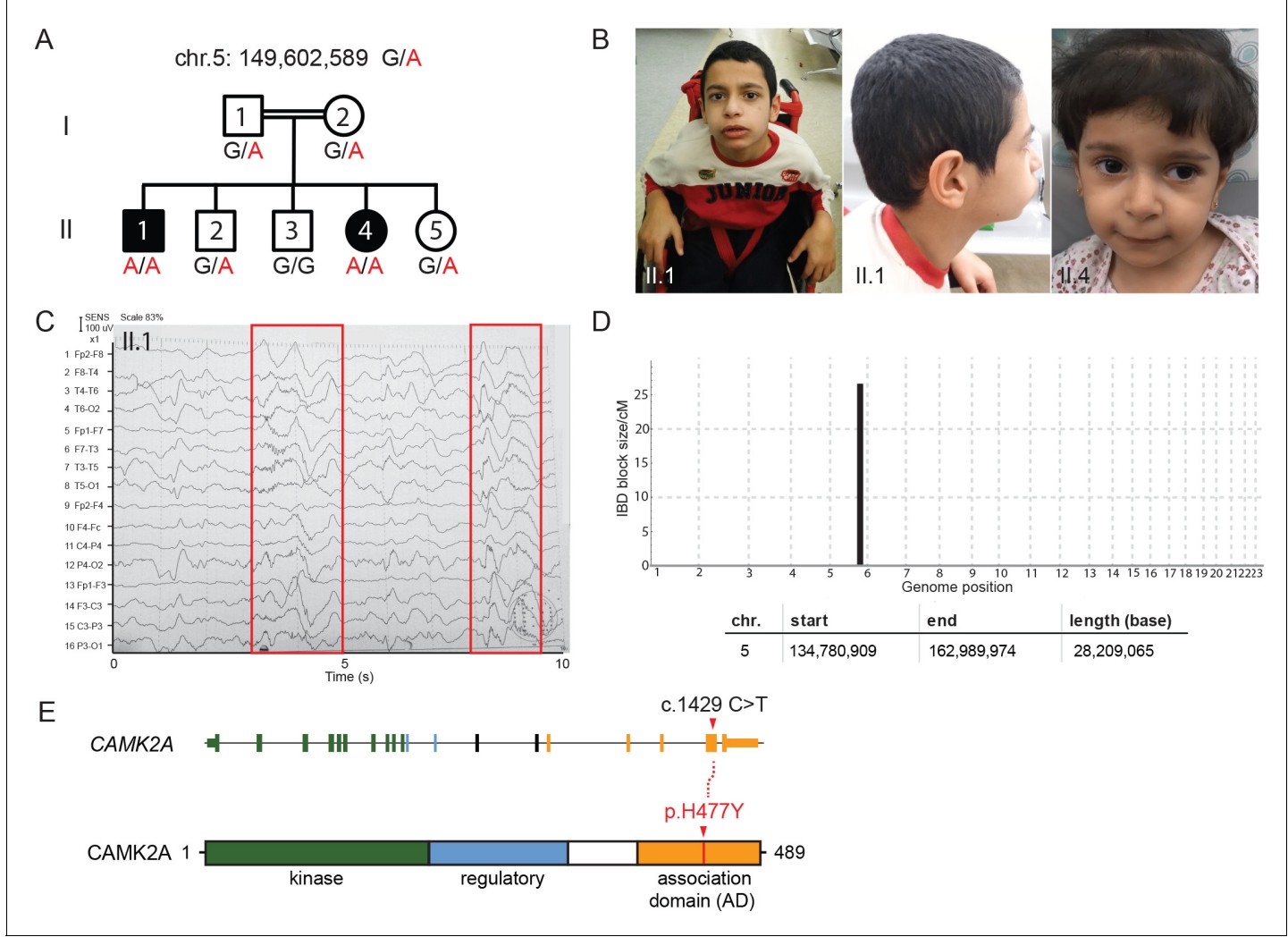

**Figure 1.** A new syndrome of global neuro-developmental delay with seizures caused by a biallelic mutation in *CAMK2A*. (**A**) Pedigree of a consanguineous Jordanian family with two affected siblings with germline homozygous mutations in *CAMK2A*. The genotypes of all individuals were verified by Sanger sequencing. (**B**) Photographs of the two affected siblings with normal head circumferences. (**C**) EEG graph of patient II.I showing abnormal epileptiform transients (red boxes) (**D**) Homozygosity mapping delineates one candidate locus on chromosome 5. (**E**) *CAMK2A* exonic structure and CAMK2A protein domains. Patients II:1 and II:4 carry biallelic missense mutation p. H477Y located in CAMK2A association domain (AD). Nucleotide change c.1429 C > T refers to position on *CAMK2A* cDNA.

DOI: https://doi.org/10.7554/eLife.32451.003

The following figure supplement is available for figure 1:

**Figure supplement 1.** Genetic and clinical findings from the two patients with global developmental delay.
DOI: https://doi.org/10.7554/eLife.32451.004

The CAMK2A enzyme consists of an N-terminal catalytic kinase domain, a $Ca^{2+}$-calmodulin-binding regulatory domain and a C-terminal association domain (AD) that is necessary for the assembly of the 12- or 14-subunit holoenzyme. The identified mutation, p.H477Y is located in the association domain (*Figure 1E*) and affects a histidine residue that is invariant across all vertebrate and invertebrate CAMK2A homologues. It is also conserved in other human CAMK2 paralogs with a similar association domain such as CAMK2B, CAMK2D and CAMK2G (*Figure 3A*, *Figure 3—figure supplement 1A*). In addition, this mutation is predicted to be deleterious by both SIFT, Poly-Phen and MCAP algorithms. Structurally, His477 is located at the dimeric interface that forms part of the extensive interaction surface at the 'equatorial' plane of the CAMK2A holoenzyme (*Figure 3B*). Based on prior findings that CAMK2A oligomerization through its association domain is indispensable for substrate phosphorylation and synaptic localization (*Bhattacharyya et al.,*

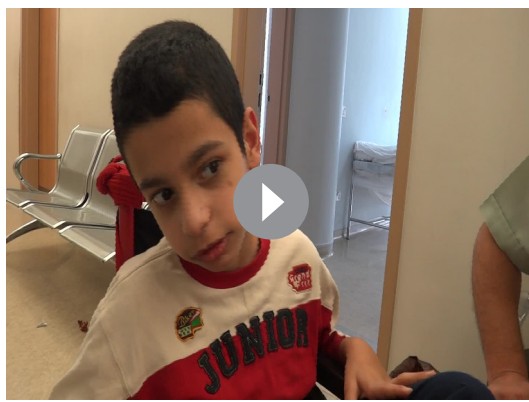

**Video 1.** Video of patient II.1
DOI: https://doi.org/10.7554/eLife.32451.005

2016), we hypothesize that the p.H477Y allele is hypomorphic and that biallelic loss-of-function in CAMK2A is the direct cause for the neurodevelopmental phenotypes in the two probands.

To measure the oligomerization potential of the CAMK2A[H477Y] mutant in cells, we transiently expressed FLAG-tagged wild-type CAMK2A and CAMK2A[H477Y] mutant in 293 T cells, which do not express endogenous CAMK2A. A third mutant CAMK2A[H477X], which lacks amino acids 477–489 and thus encodes a truncated association domain, was used as an additional control. Using native lysis conditions that preserve non-covalent macromolecular interactions, we found that wild-type CAMK2A forms a prominent complex with an apparent molecular weight ~1 MDa (*Figure 3C*, Native-PAGE, lane 2), which is consistent with the 12- or 14- subunit CAMK2A holoenzyme (*Bhattacharyya et al., 2016*). As compared to wild-type CAMK2A, the ~1 MDa, putatively oligomeric species was drastically reduced for the p.H477Y mutant and was undetectable for the p.H477X mutant (*Figure 3C*, Native-PAGE, lane 3 and 4 vs. lane 2). Next, we examined the ability of *in vitro* translated CAMK2A to self-oligomerize in a cell-free system. In contrast to the negative control protein GFP, wild-type FLAG-CAMK2A efficiently co-immunoprecipitated with HA-CAMK2A (*Figure 3D*, Lane 10 vs 11). This self-association was preserved between CAMK2A[WT] and CAMK2A[H477Y] (*Figure 3D*, lanes 12 and 15), but was completely lost between wild-type CAMK2A and the p.H477X mutant (*Figure 3D*, lane 13). In contrast, we could not detect any self-association between FLAG- CAMK2A[H477Y] and HA- CAMK2A[H477Y] (*Figure 3D*, Lane 16). Taken together, these results suggest that the missense p.H477Y partially disrupts the self-association between identical CAMK2A molecules, which had been shown to be required for holoenzyme assembly. The partial loss of function of the p.H477Y mutant, as compared to a more severe mutation p.H477X, is consistent with the observed autosomal recessive inheritance of the disease in the family, where the heterozygous carriers do not display apparent neuro-developmental symptoms.

During the course of this analysis, we noticed that both p.H477Y and p.H477X mutants had reduced protein abundance. This effect on the p.H477Y mutant was, however, subtle and could not readily explain the difference in the CAMK2A oligomer observed in the native gel (*Figure 3C*, SDS-PAGE). As it is known that in general fully assembled oligomeric complexes have enhanced stability *in vivo* compared to partially assembled complexes with disrupted folding like CAMK2A[H477Y]

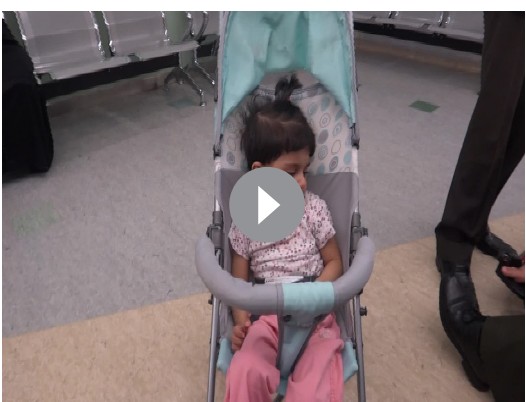

**Video 2.** Video of patient II.4
DOI: https://doi.org/10.7554/eLife.32451.006

(*Lord, 1996*; *Oromendia et al., 2012*), we hypothesized that the p.H477Y mutant may exhibit reduced stability and undergo proteasomal degradation. To test this, 293T cells were transfected with reporter constructs that encode GFP tagged wild-type and mutant CAMK2A followed by a self-cleaving peptide T2A and mCherry (*Figure 3E*). The intensity of GFP fluorescence was used as a direct quantitative measure of CAMK2A stability with the mCherry as an internal control for transfection and translational efficiency. We observed a significant reduction in GFP intensity in cells expressing CAMK2A[H477Y] and CAMK2A[H477X] as compared to wild-type CAMK2A or GFP alone (*Figure 3E*) despite comparable mCherry fluorescence levels. This reduction was rescued when we treated

**Table 1.** List of homozygous variants identified by Whole Exome Sequencing.

| Chr | Position | Gene | cDNA variant | Protein variant |
|---|---|---|---|---|
| 1 | 33,476,435 | AK2 | c.*45–1G > T | |
| 1 | 36,752,343 | THRAP3 | c.512C > T | p.Ser171Phe |
| 1 | 36,932,102 | CSF3R | c.2273C > T | p.Thr758Ile |
| 1 | 39,758,439 | MACF1 | c.1931G > T | p.Gly644Val |
| 1 | 145,365,316 | NBPF10 | c.9941G > A | p.Gly3314Glu |
| 2 | 11,758,842 | GREB1 | c.3841G > A | p.Ala1281Thr |
| 2 | 29,404,617 | CLIP4 | c.1976G > A | p.Arg659Gln |
| 2 | 64,779,195 | AFTPH | c.587G > A | p.Gly196Glu |
| 2 | 238,277,211 | COL6A3 | c.4895G > A | p.Arg1632Gln |
| 2 | 241,987,827 | SNED1 | c.1369G > A | p.Glu457Lys |
| 3 | 38,348,802 | SLC22A14 | c.574G > A | p.Ala192Thr |
| 3 | 44,672,687 | ZNF197 | c.524C > T | p.Ala175Val |
| 3 | 47,452,772 | PTPN23 | c.3484C > T | p.Arg1162Trp |
| 3 | 52,556,184 | STAB1 | c.6403C > G | p.Pro2135Ala |
| 3 | 67,426,232 | SUCLG2 | c.1235T > C | p.Ile412Thr |
| 3 | 197,422,844 | KIAA0226 | c.1366C > T | p.Arg456Trp |
| 4 | 9,174,981 | FAM90A26P | c.83T > G | p.Val28Gly |
| 4 | 9,175,603 | FAM90A26P | c.211C > G | p.Pro71Ala |
| 4 | 10,089,539 | WDR1 | c.743A > G | p.His248Arg |
| 4 | 15,529,151 | CC2D2A | c.1231T > G | p.Ser411Ala |
| 5 | 74,021,852 | GFM2 | c.1820_1825delTTGAGT | p.Glu608_Phe609del |
| 5 | 78,610,479 | JMY | c.2464C > A | p.Pro822Thr |
| **5** | **149,602,589** | **CAMK2A** | **c.1429C > T** | **p.His477Tyr** |
| 5 | 154,199,950 | C5orf4 | c.928G > A | p.Glu310Lys |
| 5 | 156,456,715 | HAVCR1 | c.1090G > A | p.Ala364Thr |
| 5 | 156,479,452 | HAVCR1 | c.590_592delCAA | p.Thr198del |
| 6 | 26,509,392 | BTN1A1 | c.1571G > A | p.Gly524Glu |
| 6 | 27,215,709 | PRSS16 | c.119G > A | p.Ser40Asn |
| 6 | 32,806,007 | TAP2 | c.4C > T | p.Arg2Trp |
| 6 | 33,260,924 | RGL2 | c.1876G > A | p.Gly626Arg |
| 6 | 38,704,952 | DNAH8 | c.221C > A | p.Ala74Asp |
| 6 | 43,412,643 | ABCC10 | c.2807C > T | p.Pro936Leu |
| 6 | 129,932,746 | ARHGAP18 | c.1054C > T | p.Arg352Ter |
| 6 | 131,946,054 | MED23 | c.235C > T | p.Leu79Phe |
| 6 | 151,674,121 | AKAP12 | c.4595_4596insGGC | p.Asp1532delinsGluAla |
| 6 | 168,479,677 | FRMD1 | c.98A > C | p.Glu33Ala |
| 7 | 5,352,665 | TNRC18 | c.7851_7856dupCTCCTC | p.Ser2618_Ser2619dup |
| 7 | 45,123,857 | NACAD | c.1922T > C | p.Val641Ala |
| 7 | 143,884,437 | ARHGEF35 | c.1040C > T | p.Thr347Ile |
| 7 | 149,422,981 | KRBA1 | c.1304C > T | p.Ala435Val |
| 7 | 151,680,130 | GALNTL5 | c.428A > G | p.Tyr143Cys |
| 8 | 12,285,064 | FAM86B1|FAM86B2 | c.310T > C | p.Ser104Pro |
| 8 | 12,285,250 | FAM86B2 | c.808C > T | p.Arg270Trp |
| 8 | 86,574,132 | REXO1L1 | c.1595A > C | p.Asp532Ala |

*Table 1 continued on next page*

Table 1 continued

| Chr | Position | Gene | cDNA variant | Protein variant |
|---|---|---|---|---|
| 9 | 12,775,863 | LURAP1L | c.149_150insTGGCGG | p.Gly49_Gly50dup |
| 9 | 40,706,047 | FAM75A3 | c.3704A > G | p.His1235Arg |
| 9 | 41,323,425 | FAM75A4 | c.1908C > T | p.Arg637Trp |
| 9 | 41,323,469 | FAM75A4 | c.1864G > A | p.Gly622Asp |
| 9 | 43,822,668 | CNTNAP3B | c.1222C > T | p.Leu408Phe |
| 10 | 51,748,684 | AGAP6 | c.209G > A | p.Arg70Gln |
| 10 | 81,471,741 | FAM22B | c.2137T > C | p.Trp713Arg |
| 11 | 1,651,198 | KRTAP5-5 | c.129_137delAGGCTGTGG | p.Gly44_Gly46del |
| 11 | 12,316,388 | MICALCL | c.1408_1410dupCCT | p.Pro470dup |
| 12 | 7,045,917 | ATN1 | c.1488_1508delGCAGCAGCAGCAGCAGCAGCA | p.Gln496_Gln502del |
| 12 | 7,045,920 | ATN1 | c.1491_1508delGCAGCAGCAGCAGCAGCA | p.Gln497_Gln502del |
| 13 | 99,461,564 | DOCK9 | c.1271_1272insA | p.Leu425LeufsTer? |
| 13 | 114,503,875 | FAM70B | c.500_509 + 72delCCTGCGGGAGG TGAGGGGCACCGGGGACCCCCATATC TACACCTGCGGGAGGTGAGGGGC GCTGGGGACCCCCGTATCTACA | |
| 14 | 105,411,514 | AHNAK2 | c.10274C > T | p.Ala3425Val |
| 14 | 106,994,118 | IGHV3-48 | c.47G > A | p.Gly16Asp |
| 16 | 29,496,359 | | c.916T > C | p.Ser306Pro |
| 16 | 30,772,988 | C16orf93 | c.82G > A | p.Ala28Thr |
| 16 | 70,215,817 | CLEC18C | c.521C > T | p.Ala174Val |
| 17 | 39,211,189 | KRTAP2-2 | c.275G > C | p.Cys92Ser |
| 19 | 1,026,716 | CNN2 | c.56A > C | p.Lys19Thr |
| 19 | 10,084,460 | COL5A3 | c.3584T > C | p.Val1195Ala |
| 19 | 14,517,213 | CD97 | c.1892G > A | p.Ser631Asn |
| 21 | 36,042,462 | CLIC6 | c.776_805delGCGTAGAAGCGGG GGTCCCGGCGGGGGACA | p.Val260_Ser269del |
| 22 | 18,834,773 | | c.329C > T | p.Thr110Ile |
| X | 48,920,059 | CCDC120 | c.110A > G | p.Asp37Gly |
| X | 55,116,478 | PAGE2 | c.25T > A | p.Ser9Thr |
| X | 150,832,702 | PASD1 | c.954_971delCCCAATGGACCAGCAGGA | p.Pro319_Asp324del |
| X | 153,050,158 | SRPK3 | c.1_5delGACAG | p.Thr2LeufsTer57 |
| X | 154,010,046 | MPP1 | c.978A > C | p.Glu326Asp |

DOI: https://doi.org/10.7554/eLife.32451.007

the cells with MG132, which blocked proteasomal degradation. MG132 treamentled to enhanced accumulation of the p.H477Y and p.H477X mutant, with a greater effect on p.H477X (*Figure 3F*, lane 3, 4 vs lane 7, 8, *Figure 3—figure supplement 1B*). By contrast, the level of the wild-type protein was reduced, likely due to the toxic effects of MG132 (*Figure 3F*, lane 2 vs. lane 6). These results suggest that in addition to causing reduced holoenzyme assembly, the p.H477Y mutation might also directly or indirectly reduce the overall CAMK2A levels by compromising its half-life.

To unequivocally demonstrate the pathogenicity of the p.H477Y allele *in vivo*, we performed rescue experiments using *C. elegans*. A single ortholog of CAMK2, encoded by *unc-43*, is present in the worm genome and its functions in the nervous system are welldocumented (*Reiner et al., 1999*). To study the neuronal defects caused by loss of *unc-43*, we focused on motor neuron, DA9, which has its cell body located in the pre-anal ganglion with a dendrite that extends anteriorly and a posteriorly oriented axon extending via a commissure into the dorsal nerve cord, where it proceeds anteriorly to form approximately 25 *en passant* synapses onto body wall muscles and

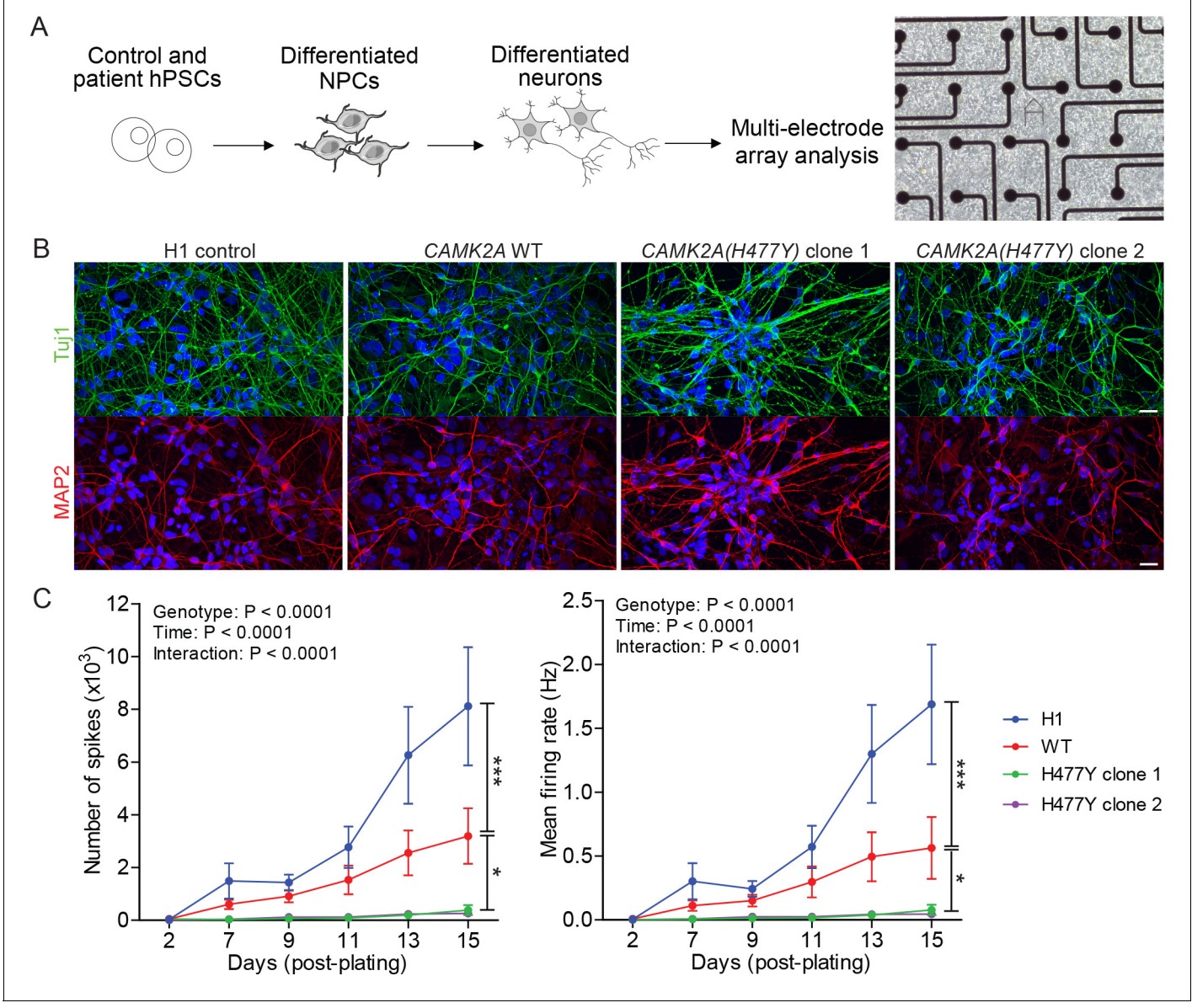

**Figure 2.** *CAMK2A* mutant iPSC-derived neurons are functionally less active. (**A**) Schematic of the hPSC-derived neuronal activity assay with representative image of iPSC-derived neurons plated on a multi-electrode array (**B**) Representative confocal images of immunofluorescence staining of neuronal lineage markers TUJ1 (green) and MAP2 (red) show efficient differentiation of iPSCs into neurons. Scale bar represents 20 μm. (**C**) Graphs showing the number of neuron-evoked spikes and mean firing rate detected by multi-electrode arrays. (n = 7 per line per time-point; Values shown as mean ±SEM; Two-way ANOVA with Tukey post-hoc test; *p<0.05 and ***p<0.001).
DOI: https://doi.org/10.7554/eLife.32451.008

reciprocal inhibitory neurons (*Figure 4A*) (*Klassen and Shen, 2007*; *Maeder et al., 2014*). Using mCherry-tagged UNC-43, we observed that UNC-43 could be found along the entire DA9 neuron but concentrated at synaptic boutons (*Figure 4B*). The homologous patient-specific mutation p. H466Y in UNC-43 (homologous to p.H477Y in human CAMK2A) significantly disrupted the synaptic localization of UNC-43 and caused the protein to be dispersed throughout the entire axon (*Figure 4B*). To test the consequence of UNC-43 mutation on DA9 synaptic function, we used the *itr-1 pB* promoter to express the fluorescently conjugated synaptic vesicle marker RAB-3 (RAB3:: GFP) within DA9 in both the wild-type and *unc-43* mutant background. In wild-type animals, RAB-3::GFP accumulated in discrete puncta along the axon of DA9 at stereotyped synaptic locations.

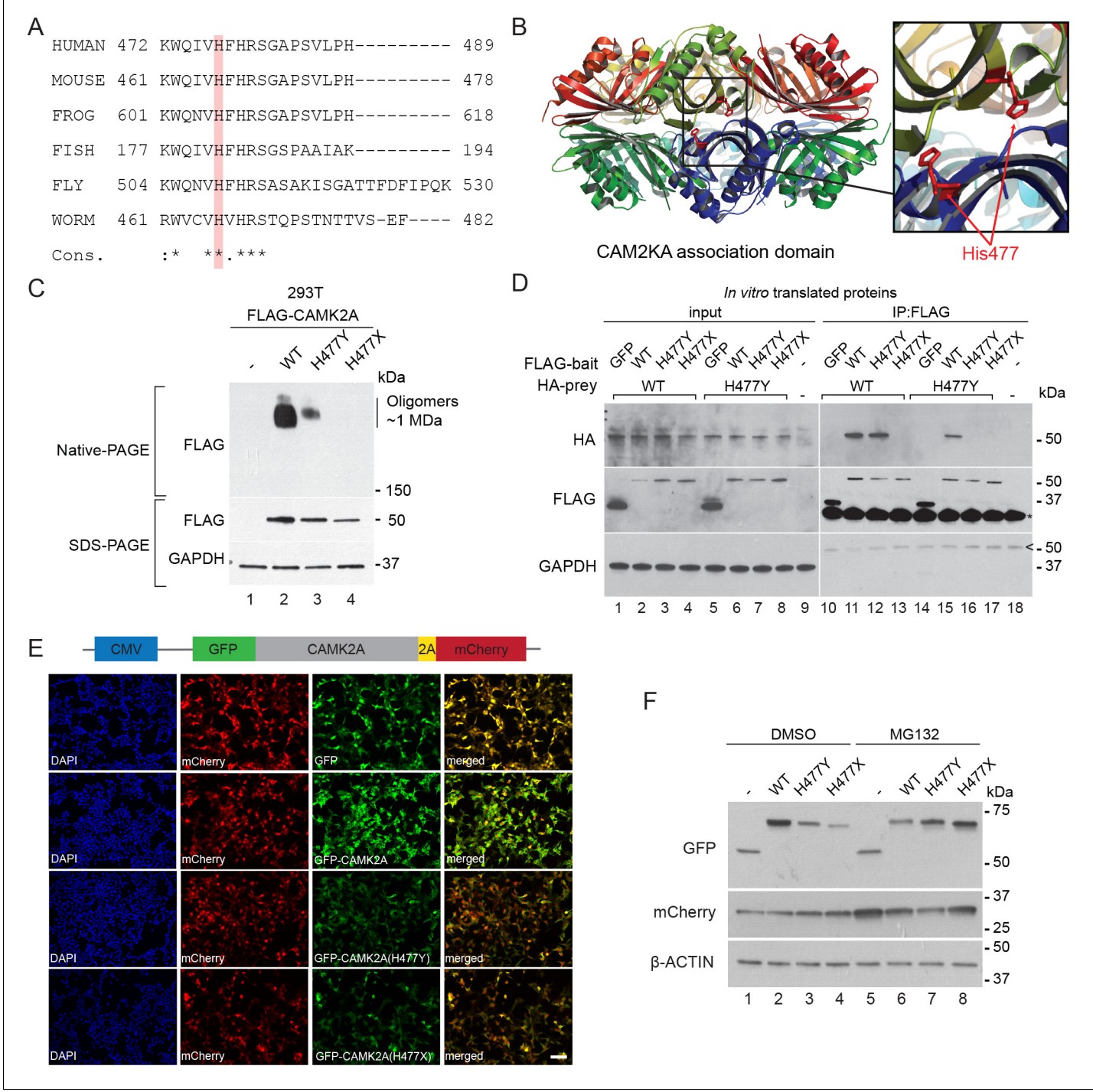

**Figure 3.** p.H477Y affects CAMK2A oligomerization and protein stability. (**A**) Sequence conservation of CAMK2A homologs. Histidine 477 (H477) is highlighted in red. (**B**) X-ray crystal structure of human CAMK2A AD tetradecamer (PDB: 5IG3). H477 (red) is located at the equatorial dimer interface. (**C**) Defective oligomerization of the p.H477Y mutant. 293 T cells were transiently transfected with FLAG tagged wild-type CAMK2A and CAMK2A[H477Y]. A third mutant, CAMK2A[H477X] which lacks part of the AD (a.a. 478–489) was used as positive control. (**D**) Defective self-association of the p.H477Y mutant. The indicated FLAG- and HA-tagged CAMK2A wild-type and mutant proteins were synthesized *in vitro* using rabbit reticulocyte lysate. FLAG-GFP was used a negative control. FLAG-tagged proteins were immunoprecipitated using anti-FLAG M2 agarose resin in the presence of 1% NP40. Co-immunoprecipitated proteins were analyzed by SDS-PAGE. *, IgG light chain.ˆ, IgG heavy chain. (**E**) p.H477Y mutation lowers expression of CAMK2A in cells. 293 T cells were transfected with reporter plasmids encoding GFP-tagged wild-type CAMK2A, CAMK2A[H477Y] and CAMK2A[H477X] mutants, followed by T2A peptide and mCherry. Representative confocal images show lower expression of mutant GFP- CAMK2A[H477Y] compared to wild-type.

*Figure 3 continued on next page*

*Figure 3 continued*

Scale bar represents 100 μm. (**F**). p.H477Y decreases CAMK2A stability via proteasomal degradation. 293 T cells were transfected as in (**E**) and treated with 10 μM MG132 or DMSO for 16 hr. 10 μg total cell lysate was used for SDS-PAGE and Western blot.
DOI: https://doi.org/10.7554/eLife.32451.009
The following figure supplement is available for figure 3:

**Figure supplement 1.** Decreased stability and defective cytoplasmic localization of the CAMK2A[H477Y] mutant.
DOI: https://doi.org/10.7554/eLife.32451.010

In the canonical *unc-43(e408)* loss-of-function mutant, we observed a reduction in individual pre-synaptic puncta fluorescence intensity as compared to wild-type animals. This phenotype could be rescued by cell-autonomous expression of wild-type *unc-43* in DA9. However, expression of *unc-43* harboring the homologous patient mutation UNC-43[H466Y] failed to rescue this defect (*Figure 4C,D*). In addition, transgenic expression of human wild-type CAMK2A fully rescued this defect, confirming the high degree of functional conservation between CAMK2A homologs, while the patient-derived human CAMK2A[H477Y] failed to do so (*Figure 4C*, bottom panels). These results suggest that the p.H477Y mutation is defective in supporting pre-synaptic function in *C. elegans*.

Immediately posterior to the stretch of presynaptic puncta is an asynaptic domain within the DA9 axon that is devoid of any RAB-3::GFP fluorescence in wild-type animals (*Figure 4E*, top panel). Loss of *unc-43* results in the mislocalization of the synaptic marker RAB3::GFP into this asynaptic region (*Figure 4E*, bottom panel). This defect could also be rescued by cell autonomous expression of either *unc-43* or human *CAMK2A*. Patient derived mutation CAMK2A[H477Y] or the worm homologous mutation UNC-43[H466Y] both failed to rescue this phenotype (*Figure 4E*). In addition, expression of UNC-43[H466Y] in wild-type animals did not cause any synaptic defect or mis-localizaton of RAB3::GFP, suggesting that the mutation does not have dominant negative effects (*Figure 4F*).

We further tested if the patient derived mutation in CAMK2A impacted worm locomotor behavior. Null mutants for *unc-43* are flaccid in posture and move with a flattened uncoordinated waveform. The animals are variably convulsive, often spontaneously contracting and relaxing their body-wall muscles in brief repeating bursts that resemble seizures (*Reiner et al., 1999*). We expressed either wild-type UNC-43 or UNC-43[H466Y] in the muscles and neurons of *unc-43(e408)* mutant worms and scored the behavior of young adults in a double-blind experiment. Only the wildtype UNC-43 was able to rescue the movement defects in *unc-43(e408)* animals, but not UNC-43[H466Y], suggesting that UNC-43[H466Y] is not functional (*Figure 4G*). Together, the data show that CAMK2A[H477Y] is a loss-of-function mutation which fails to support synaptic function in vivo.

In summary, we have identified an autosomal recessive neurodevelopmental syndrome characterized by growth delay, frequent seizures and severe intellectual disability that is caused by a biallelic germline loss-of-function mutation in CAMK2A. Mechanistically this mutation disrupts CAMK2A self-oligomerization and holoenzyme assembly via its association domain. Our functional results are consistent with the high degree of evolutionary conservation of the affected residue H477 in CAMK2A orthologs, as well as previous structural data demonstrating that His477 is located in the interface between two stacked CAMK2A subunits, which together form the basic repeat unit of the ring-shaped holoenzyme (*Bhattacharyya et al., 2016*; *Stratton et al., 2014*). The pathogenicity of the biallelic p.H477Y mutation is furthered highlighted by rescue experiments in *C. elegans*, where in contrast to wild-type human CAMK2A the p.H477Y mutant failed to rescue the neuronal and behavioral defect in *unc-43* (CAMK2 ortholog) null worms. Together, these data demonstrate that the loss of function of CAMK2A is the most plausible genetic cause for the neurodevelopmental defects observed in the two affected siblings.

## Discussion

CAMK2 plays important and evolutionary conserved roles in synaptic plasticity, neuronal transmission and cognition in near all model organisms examined, and several groups have shown that somatic mutations in human CAMK2 isoforms may contribute to neurological disorders (*Ghosh and*

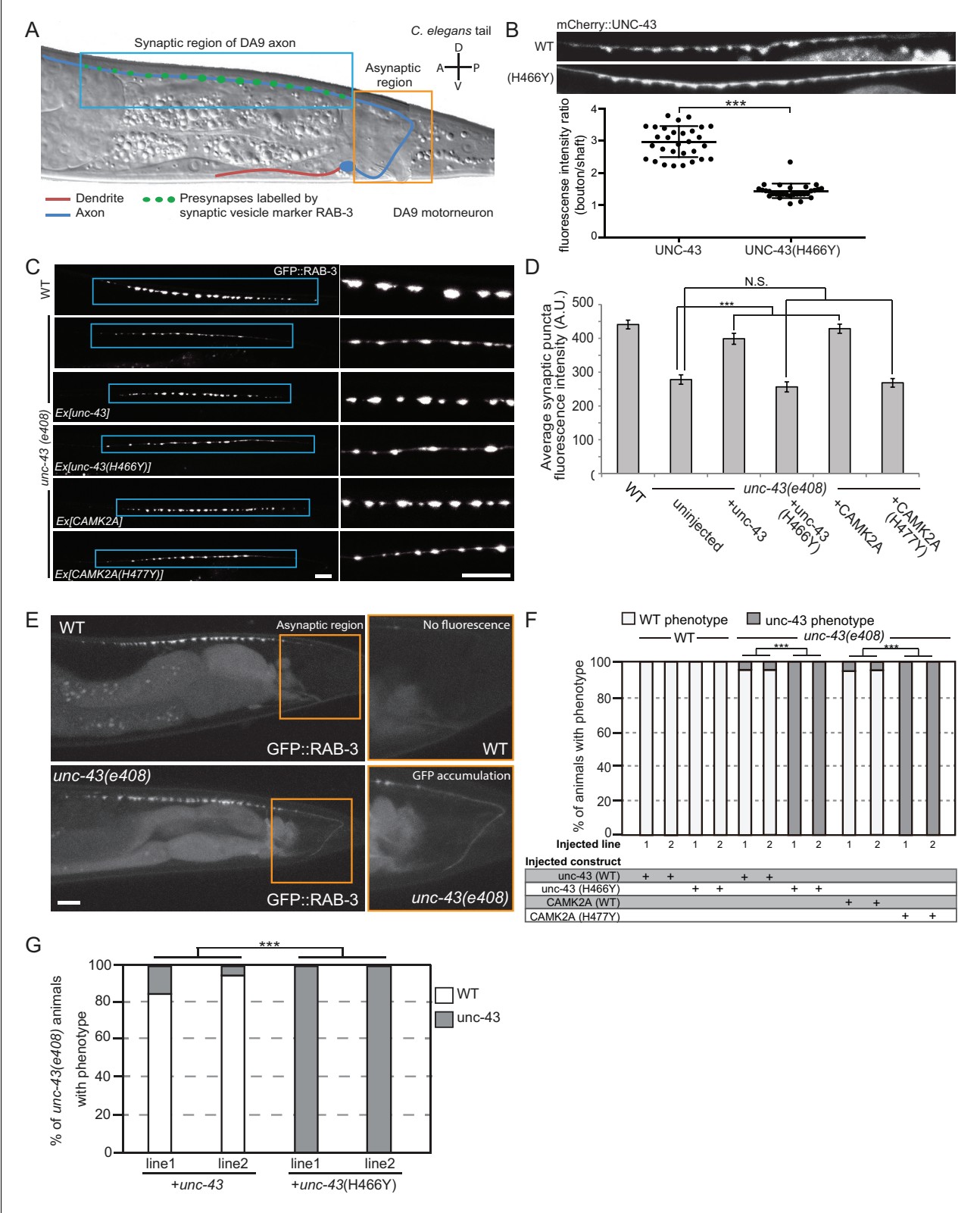

**Figure 4.** CAMK2A[H477Y] mutant fails to rescue synaptic defects in *unc-43 C. elegans* neurons. (**A**) Schematic drawing of *C. elegans* motor neuron, DA9 in the tail region. DA9 extends a dendrite (red) anteriorly and an axon (blue) that extends posteriorly crosses the midline of the animal and anteriorly in the dorsal nerve cord (DNC). It forms approximately 20 *en passant* synapses within a discrete stretch along the DNC (blue box). DA9 presynaptic vesicles were marked with RAB-3 (GFP::RAB-3). The asynaptic region (yellow box) is devoid of any synaptic vesicle accumulation. (**B**) The localization of
*Figure 4 continued on next page*

*Figure 4 continued*

mCherry::UNC-43 and mCherry::UNC-43(H466Y) in DA9 synapses. Note that UNC-43 accumulates at synaptic boutons while UNC-43(H466Y) is diffusely localized. Fluorescent intensity of mCherry::UNC-43 was measured at synaptic boutons and along the axonal shaft. Graph plots the ratio of fluorescence intensity at synaptic boutons compared to the axonal shaft of 30 synapses from three animals. Graph shows the mean and error bars show SEM, ***p-value $6.32e^{-19}$, Student's T-test. (C) Representative confocal images demonstrating presynaptic puncta size changes between WT and *unc-43(e408)* mutants. *unc-43* mutants have smaller presynaptic puncta along the DNC. This defect is rescued by expression of either UNC-43 or CAMK2A in DA9 whilst the mutated UNC-43^H466Y and CAMK2A^H477Y fail to rescue. (D) Quantification of average puncta intensity from WT and *unc-43(e408)* animals. Error bars represent SEM with number of synaptic puncta quantified n > 80, N.S. is not significant, ***p-value<0.001 (uninjected vs *unc-43* p-value $5.0e^8$, uninjected vs *unc-43*^H466Y p-value 4.17, uninjected vs CAMK2A p-value $4.25e^{-12}$, uninjected vs *CAMK2A*^H477Y p-value 9.40), One-Way ANOVA with Bonferroni posthoc test. (E) Representative confocal images showing mislocalization of GFP::RAB-3 into the asynaptic region (yellow box) in *unc-43* DA9 neuron. (F) Rescue of the *unc-43(e408)* phenotype by DA9 cell-specific expression of UNC-43 or CAMK2A. UNC-43^H466Y and CAMK2A^H477Y fail to rescue the *unc-43* phenotype. Graph shows the percentage of animals with the WT and *unc-43* mutant phenotypes. ***p<0.001 (*unc-43* vs *unc-43*^H466Y p-value $2.13e^{-51}$, CAMK2A vs CAMK2A^H477Y p-value $3.77e^{-50}$), Fisher Exact test with n = 100 animals scored for each line. (G) Behavioral rescue by expressing wild-type UNC-43 or UNC-43^H466Y in *unc-43(e408)* mutants. The behavior was scored as either wild-type or unc-43. Two independent worm lines were analyzed for each condition. *** p-value $5.29e^{-41}$, Fisher Exact test with n = 100 animals scored for each line.

DOI: https://doi.org/10.7554/eLife.32451.011

*Giese, 2015*; *Robison, 2014*; *Takemoto-Kimura et al., 2017*). Notably, a de novo p.E183V mutation in the CAMK2A catalytic domain was shown to cause autism spectrum disorder (*Stephenson et al., 2017*). This mutation was shown to act in a dominant-negative manner to reduce wild-type CAMK2A auto-phosphorylation and localization to dendritic spines. While this manuscript was under revision, S. Küry *et al.* reported multiple families with intellectual disability caused by de novo, heterozygous mutations in both CAMK2A and CAMK2B kinase and auto-regulatory domains, which disrupted CAMK2 phosphorylation and caused neuronal migratory defects in murine models (*Küry et al., 2017*).

Our discovery of a novel neuro-developmental syndrome caused by biallelic CAMK2A mutations further broadens the spectrum of human neurological disorders caused by the CAMK2 family of kinases. To the best of our knowledge, this represents the first Mendelian human disease caused by biallelic CAMK2A mutations. Our functional characterization of the novel mutation p. H477Y *in vitro* and *in vivo* also reveal novel insights on how the CAMK2A holoenzyme regulates neuronal function. In contrast to all previously reported mutations in CAMK2A in intellectual disability syndromes, the p.H477Y is located within the C-terminal association domain and results in a partial but significant disruption of self-oligomerization, suggesting that the assembly of CAMK2A oligomers, in addition to its kinase function is required for neuronal function. Interestingly, the CAMK2A^H477Y mutant retains the ability to interact with wild-type CAMK2A but not with itself (*Figure 3D*). CAMK2A displays very specific cellular and subcellular expression patterns that is important for regulating substrate phosphorylation in cells (*Liu and Murray, 2012*; *Tsui et al., 2005*). The p.H477Y missense affects the subcellular localization in neurons and this may affect its ability to function efficiently. These biochemical results provide a mechanistic basis for the autosomal recessive nature of the disease in our family: the p.H477Y allele is hypomorphic and becomes pathogenic when recessively inherited in the homozygous state. We speculate that heterozygous carriers retain sufficient CAMK2A activity for proper neuronal function. As compared to the milder disease phenotypes reported by Kury et. al., the symptoms afflicting the p.H477Y patients likely represent the most severe manifestation of the CAMK2A dysfunction in humans.

Due to the high degree of conservation of CAMK2A across evolution, we employed the established *C. elegans unc-43* mutant to prove the pathogenicity of the p.H477Y mutation. UNC-43 is the worm homologue of vertebrate CAMK2. We demonstrated wild-type human CAMK2A could rescue the locomotive defects of the *unc-43* mutants, while the p.H477Y mutant failed to do so, likely due to its inability to localize into neuronal synapses (*Figure 4*). We anticipate this *in vivo* functional assay in *C. elegans* to be widely applicable to assess the pathogenicity of newly discovered CAMK2 alleles found in human diseases.

# Materials and methods

## Key resources table

| Reagent type (species) or resource | Designation | Source or reference | Identifiers | Additional information |
|---|---|---|---|---|
| Gene (human) | CAMK2A Calcium/Calmodulin Dependent Protein Kinase II Alpha | Uniprot: Isoform B (identifier: Q9UQM7-2) | Q9UQM7-2 | |
| Gene (C. elegans) | unc-43 Calcium/Calmodulin -Dependent Protein Kinase type II | Protein UNC-43 isoform d (Wormbase CDS K11E8.1d) | K11E8.1d | |
| Strain, strain background (C. elegans) | Worm (C. elegans) N2 Bristol Strain;unc-43(e408) | Caenorhabditis Genetics Center (CGC) PMID: 17941711 | 17941711 | |
| Recombinant DNA | pCDH-CMV-MCS-EF1α-Neo | Systems Biosciences (SBI) | CD514B-1 | |
| Recombinant DNA | pSM vector (a derivative of pPD49.26 with additional cloning sites) | Modified for this paper | NA | ADDGENE https://media.addgene.org/cms/files/Vec95.pdf |
| Recombinant DNA | pCDH-CMV-CAMK2A-T2A-mCherry | This paper; based on pCDH-CMV-MCS-EF1α-Neo | NA | |
| Recombinant DNA | pMIG-hOCT4 | Addgene | Plasmid #17225 | |
| Recombinant DNA | MSCV h c-MYC IRES GFP | Addgene | Plasmid #18119 | |
| Recombinant DNA | pMIG-hKLF4 | Addgene | Plasmid #17227 | |
| Recombinant DNA | pMIG-hSOX2 | Addgene | Plasmid #17226 | |
| Cell line (human) | Patient derived iPS neurons | This paper | NA | |
| Cell line (human) | 293T | Lab stock | NA | |
| Chemical compound, drug | MG132 | Sigma-Aldrich | M7449 | |
| Commercial kit | NativeMark PAGE | ThermoFisher | BN1002BOX | |
| Commercial kit | TnT Quick Coupled Transcription/Translation System | Promega | L1170 | |
| Antibody | Anti-FLAG Clone M2 | Sigma-Aldrich | F3165 | |
| Antibody | Anti-HA Clone Y-11 | Santa Cruz Biotechnology | sc-7392 | |
| Antibody | Anti-GADPH | Santa Cruz Biotechnology | sc-47724 | |
| Antibody | Anti-Tuj1 | Covance Research | MMS-435P | |
| Antibody | Anti-MAP2 | Synaptic Systems | 188 004 | |
| Cell culture reagent | 20% Knock Out Serum Replacement | Thermo Fisher | 10828–028 | |
| Cell culture reagent | bFGF | Stemgent | 37316 | |
| Cell culture reagent | Matrigel Basement Membrane Matrix | Corning | 354234 | |
| Cell culture reagent | mTeSR1 | STEMCELL Technologies | 85850 | |
| Cell culture reagent | CytoTune-iPS 2.0 Sendai Reprogramming Kit | ThermoFisher | A16517 | |
| Cell line (human) | H1 embryonic stem cells | Gift from Dr. Lawrence W. Stanton, WiCell | RRID:CVCL_C813 | |
| Antibody | Alexa Fluor 594 secondary Ab | ThermoFisher | Cat# A-11076, RRID:AB_2534120 | |
| Antibody | Alexa Fluor 488 secondary Ab | ThermoFisher | Cat# A-11001, RRID:AB_2534069 | |
| Assay system/kit | Maestro MEA System | Axion Biosystem | - | |

## Disease diagnosis and informed consent

Patients were identified and diagnosed by clinical geneticists at King Hussein Medical Centre, Amman, Jordan. Informed consent was obtained from the families for genetic testing in accordance with approved institutional ethical guidelines (Institute of Medical Biology, Singapore, A*STAR, Singapore, NUS-IRB reference code 10–051). For patients in *Figure 1*, informed consent to publish photographs and videos was obtained from parents. Genomic DNA samples were isolated from saliva using the Oragene DNA collection kit (OG-500, DNAGenotek) and a punch skin biopsy was taken from patient II.4.

## Whole exome sequencing

Whole exome sequencing of proband II-1 was performed using the Illumina TruSeq Exome Enrichment Kit for exome capture using 1 ug of genomic DNA. Illumina HiSeq2000 High-output mode was used for sequencing as 100 bp paired-end runs at the UCLA Clinical Genomics Centre and at the UCLA Broad Stem Cell Research Centre as previously described (*Hu et al., 2014*). An average coverage of 34X was achieved across the exome with 87% of these bases covered at $\geq$10X. After filtering, a total of 73 homozygous, 125 compound heterozygous and 493 heterozygous variants were protein-changing variants with population minor allele frequencies < 1%.

## Homozygosity mapping

Both parents and their five children were genotyped using Illumina Humancore-12v1 BeadChips following manufacturer's instructions. Call rates were above 99%. Gender and relationships were verified using Illumina BeadStudio. Mapping was performed by searching for shared regions that are homozygous and identical-by-descent (IBD) in the two affected children using custom programs written in the Mathematica (Wofram Research, Inc.) data analysis software (Source code file 1 - IBD linkage program). Candidate regions were further refined by exclusion of common homozygous segments with any unaffected family members. The confidence criteria to identify IBD blocks were a minimum of 5 cM. We identified one shared candidate loci on chromosome 5.

## Cell culture and plasmid transfection

HEK 293 T cells (ATCC Cat# CRL-3216, RRID:CVCL_0063, from Lab Stock) were cultured in DMEM media (Gibco) supplemented with 10% FBS. Cell line identity was authenticated by commercial human STR profiling with ATCC (ATCC, #135-XV). All cell lines used in this paper tested negative for mycoplasma using Lonza MycoAlert (Lonza LT07). For transient transfection, $6 \times 10^5$ cells per well were seeded in 6 well plates 24 hr before being transfected with Lipofectamine 2000 (Thermo-Fisher)-complexed plasmids in OPTIMEM (ThermoFisher). To construct the CAMK2A expression plasmids, human CAMK2A cDNA was PCR-amplified from ImageClone with AscI and PacI restriction sites and cloned into pCDNA3.1 with an N-terminal 3xFLAG or 3xHA tag. All CAMK2A mutants were generated using QuikChange XL (Agilent). Cells were treated with MG132 (Sigma, M8699) at 5 µM to block proteasome degradation.

## Generation of iPSCs

### Control iPSCs from an unrelated but ethnically and sex-matched individual

Fibroblasts were transduced with *OCT4*, *SOX2*, *KLF4* and *c-MYC* (Addgene plasmids #17225, #17226, #17227 were gifts from George Daley, and #18119 a gift from John Cleveland) as previously described (*Park et al., 2008*). After 4 days, transduced cells were reseeded onto irradiated mouse embryonic fibroblast in human ES cell medium (DMEM/F12 (Sigma, D6421) supplemented with 20% Knock Out Serum Replacement (Thermo Fisher Scientific, 10828–028), 0.1 mM 2-mercaptoethanol (Thermo Fisher Scientific, 21985–023), 2 mM L-glutamine (Thermo Fisher Scientific, 25030), 0.2 mM NEAA (Thermo Fisher Scientific, 11140–050) and 5 ng/mL bFGF (Stemgent, 03–0002). iPSC colonies were picked between days 17–28 and maintained in matrigel (Corning, 354234) and mTeSR1 (STEM-CELL Technologies, 85850) for expansion.

### Patient-derived iPSC from proband, II.4

Dermal primary fibroblasts were reprogrammed using the CytoTune-iPS 2.0 Sendai Reprogramming Kit (Thermo Fisher Scientific, A16517) in accordance with the manufacturer's instructions. Cells were

passaged and plated onto irradiated mouse embryonic feeders 7 days after viral transfection in human ES cell medium (DMEM/F12 (Sigma, D6421) supplemented with 20% Knock Out Serum Replacement (Thermo Fisher Scientific, 10828–028), 0.1 mM 2-mercaptoethanol (Thermo Fisher Scientific, 21985–023), 2 mM L-glutamine (Thermo Fisher Scientific, 25030), 0.2 mM NEAA (Thermo Fisher Scientific, 11140–050) and 5 ng/mL bFGF (Stemgent, 03–0002). iPSC colonies were picked between days 17–28 and maintained in Matrigel Basement Membrane Matrix (Corning, 354234) and mTeSR1 (STEMCELL Technologies, 85850) for expansion.

## Neuronal differentiation and Multi-electrode Array Recordings

iPSCs-derived NPCs were differentiated into neurons for 21 days using a previously published protocol (*Xu et al., 2017a*). Briefly, NPCs were plated at a density of 50,000 cells/cm$^2$ in a poly-L-ornithine and laminin-coated plates, cultured in N2B27 medium supplemented BDNF (20 ng/ml), GDNF (20 ng/ml), cAMP (N6,2'-O-dibutyryladenosine 3',5'-cyclic monophosphate; Sigma; 0.3 mM) and ascorbic acid (0.2 mM). H1 embryonic stem cells (Gift from Lawrence W. Stanton, WiCell, RRID:CVCL_C813) were also differentiated into neurons to use as controls. H1 embryonic stem cells were verified by karyotyping (Cytogenetics Laboratory, KK Women's and Children's Hospital, Singapore) and checked for pluripotency by differentiation into the three germ layers marked by Nestin (ectoderm), AFP (endoderm) and ASM-1 (mesoderm). For the immunofluorescence staining, neurons were fixed for 15 min in ice cold 4% (w/v) paraformaldehyde. Permeabilization using 0.3% (v/v) Triton-X in 1X PBS was performed for 10 min then incubated with 1:1000 mouse anti-Tuj1 (Covance Research Products Inc Cat# MMS-435P, RRID:AB_2313773), 1:1500 guinea pig anti-MAP2 (Synaptic Systems Cat# 188 004, RRID:AB_2138181) overnight at 4°C in 5% (w/v) BSA diluted with 1X PBS. For visualization, 1:1000 secondary antibody conjugated to Alexa Fluor 594 (Thermo Fisher Scientific Cat# A-11076, RRID:AB_2534120) or Alexa Fluor 488 (Thermo Fisher Scientific Cat# A-11001, RRID:AB_2534069) was applied. Counter staining for nuclei were performed using Dapi. Images were captured using the FV3000 Olympus confocal.

For the multi-electrode array (MEA) recordings, neurons on day 21 were dissociated and replated on 0.1 polyethylenimine (Sigma)-coated 48 well MEA plates (Axion Biosystems) in BrainPhys media supplemented with BDNF, GDNF, cAMP and ascorbic acid as previously described (*Xu et al., 2017b*). Spontaneous neuronal activity was observed and recorded at 37°C for 5 min every 2–3 days using the Maestro MEA System (Axion Biosystem). Independent measurements were taken from seven wells for each condition (technical replicates).

## *In vitro* transcription/translation and co-immunoprecipitation

CAMK2A proteins were synthesized *in vitro* using TNT T7 Quick Coupled Transcription/Translation (Promega) with 1 µg of plasmids in 20 µl reaction volumes for 90 mins at 30°C. For co-immunoprecipitation, the reactions were diluted 10x in TBS (100 mM Tris-HCl, pH = 8, 150 mM NaCl) with 1% Nonidet P40 (NP40) and incubated with 10 µl anti-FLAG M2 agarose beads (Sigma) at 4°C overnight. Proteins were eluted with 1x Laemmli buffer after 3 washes in the 1xTBS with 1% NP40.

## Protein electrophoresis and immunoblotting

Total protein lysate was quantified using a standard Bradford assay and 10 µg of lysate was used for immunoblotting experiments. For Blue Native PAGE, cells were lysed in 1x Sample Preparation buffer (ThermoFisher) containing 1% digitonin. 1% SDS was supplemented for SDS-PAGE. All proteins were transferred to PVDF membranes using TurboBlot (Bio-rad) at 2.5 mA for seven mins. The primary antibodies used were anti-FLAG (M2, Sigma-Aldrich Cat# F3165, RRID:AB_259529), anti-GAPDH (Santa Cruz Biotechnology Cat# sc-47724, RRID:AB_627678) and anti-HA (Y-11, Santa Cruz Biotechnology Cat# sc-805, RRID:AB_631618). The secondary antibodies were anti-rabbit IgG-HRP (Jackson ImmunoResearch Labs Cat# 111-035-003, RRID:AB_2313567), anti-mouse IgG-HRP (light chain specific) (Jackson ImmunoResearch Labs Cat# 205-032-176, RRID:AB_2339056) and anti-mouse IgG-HRP (Jackson ImmunoResearch Labs Cat# 115-035-003, RRID:AB_10015289).

## Worm strains

All strains were maintained at 20°C on OP50 *E. coli* nematode growth medium plates as described (*Brenner, 1974*). N2 Bristol strain worms (WB Cat# CB4852, RRID:WB-STRAIN:CB4852) were used

as the WT reference, and the *unc-43(e408)* mutant was used. To visualize synaptic vesicles in DA9 neuron, *wyIs85* [Pitr-1::GPF::RAB-3] was used (*Klassen and Shen, 2007*).

## Transgenic lines

OTL70 *wyIs85*[Podr-1::dsred, Pitr-1::gfp::rab-3]; *jpnEx40*[Podr-1::gfp, Pmig-13::unc-43]

OTL71 wyIs85; *jpnEx41*[Podr-1::gfp, Pmig-13::unc-43]

OTL72 *unc-43(e408); wyIs85; jpnEx44*[Podr-1::gfp, Pmig-13::unc-43(H466Y)]

OTL73 *unc-43(e408); wyIs85; jpnEx45*[Podr-1::gfp, Pmig-13::unc-43(H466Y)]

OTL74 *unc-43(e408); wyIs85; jp*nEx42[Podr-1::gfp, Pmig-13::unc-43]

OTL75 *unc-43(e408); wyIs85; jpnEx43*[Podr-1::gfp, Pmig-13::unc-43]

OTL76 *unc-43(e408); wyIs85; jpnEx47*[Podr-1::gfp, Pmig-13::*CAMK2A*]

OTL77 *unc-43(e408); wyIs85; jpnEx48*[Podr-1::gfp, Pmig-13::*CAMK2A*]

OTL78 *unc-43(e408); wyIs85; jpnEx49*[Podr-1::gfp, Pmig-13::*CAMK2A*$^{H477Y}$]

OTL79 *unc-43(e408); wyIs85; jpnEx50*[Podr-1::gfp, *Pmig-13::CAMK2A*$^{H477Y}$]]

OTL82: jpnEx54[Podr-1::gfp, Pmig-13::mcheery::unc-43]

OTL83: jpnEx55[Podr-1::gfp, Pmig-13::mcherry::unc-43]

OTL84: jpnEx56[Podr-1::gfp, Pmig-13::mcherry::unc-43(H466Y)]

OTL85: jpnEx57[Podr-1::gfp, Pmig-13::mcherry::unc-43(H466Y)]

OTL86: unc-43(e408); jpnEx58[Punc-122::dsred, Punc-104::mcherry::unc-43, Phlh-1::mcherry::unc-43]

OTL87: unc-43(e408); jpnEx59[Punc-122::dsred, Punc-104::mcherry::unc-43, Phlh-1::mcherry::unc-43]

OTL88: unc-43(e408); jpnEx60[Punc-122::dsred, Punc-104::mcherry::unc-43(H466Y), Phlh-1::mcherry::unc-43(H466Y)]

OTL89: unc-43(e408); jpnEx61[Punc-122::dsred, Punc-104::mcherry::unc-43(H466Y), Phlh-1::mcherry::unc-43(H466Y)]

## Plasmids for transgenic expression in worms

Expression plasmids for transgenic worm lines were made using the pSM vector (C. Bargmann), a derivative of pPD49.26 (A. Fire). The *mig-13* promoter was cloned between SphI/AscI, and C.elegans *unc-43* isoform d or human *CAMK2α* was cloned between NheI/KpnI or AscI/NheI, respectively. P. H466Y and p.H477Y mutations were introduced by PCR-based mutagenesis using KOD-plus- high fidelity DNA polymerase (TOYOBO, Tokyo, Japan). Transgenic worms were generated as described (*Mello, 1995*). Plasmids were injected into animals at 10 ng/µl (in the case of Pmig-13::unc-43) and 50 ng/µl (in the case of Pmig-13::*CAMK2α*) together with coinjection markers at 90 ng/µl.

## Fluorescence quantification and confocal imaging

All fluorescence images of DA9 were taken in live worms immobilized with 5% agar pad, 10 µM levamisol (Sigma) and 0.1 mm polystylene beads (Polysciences, Inc., Warrington, PA, USA) with a 63×/1.4 NA objective on a Zeiss Axioplan 2 Imaging System or a Plan-Apochromat 63×/1.3 objective on a Zeiss LSM710 confocal microscope using similar imaging parameters for the same marker across different genotypes. Fluorescence quantification was determined using Image J software (ImageJ, RRID:SCR_003070).

## Behavioral analysis

*mcherry::unc-43* or *mcherry::unc-43(H466Y)* were expressed in *unc-43*(*e403*) mutant worms using both *unc-104* promoter (neuron-specific promoter) and *hlh-1* promoter (muscle-specific promoter). Two independent lines were established and analyzed. The behavioral phenotype of the transgenic worms was scored in a double-blind manner using a stereo microscope (SZX16, Olumpus, Tokyo, Japan). From the movement behavior on OP50 feeder NGM plates, each worm was classified to behave like '*wild-type*' or '*unc-43*'. 100 worms were observed for each genotype on the same day from two independently derived transgenic lines.

## Acknowledgements

We would like to thank all patient family members for their kind participation in this study. We would like to acknowledge funding support from the Strategic Positioning Fund for Genetic Orphan Diseases from the Agency for Science, Technology, and Research in Singapore. We are grateful to all members of the BR Laboratory for their support. MAP is supported by grants from the Agency for Science Technology and Research (Singapore), and the National University of Singapore (Singapore). BR is a fellow of the Branco Weiss Foundation and a recipient of the A*STAR Investigatorship and EMBO Young Investigator.

## Additional information

### Funding

| Funder | Grant reference number | Author |
|---|---|---|
| Agency for Science, Technology and Research | GODAFIT Strategic Positioning Fund | Bruno Reversade |

The funders had no role in study design, data collection and interpretation, or the decision to submit the work for publication.

### Author contributions

Poh Hui Chia, Conceptualization, Resources, Data curation, Supervision, Funding acquisition, Visualization, Methodology, Writing—original draft, Project administration, Writing—review and editing; Franklin Lei Zhong, Shinsuke Niwa, Conceptualization, Data curation, Formal analysis, Validation, Investigation, Visualization, Methodology, Writing—original draft, Project administration, Writing—review and editing; Carine Bonnard, Resources, Formal analysis, Validation, Investigation; Kagistia Hana Utami, Data curation, Validation, Project administration; Ruizhu Zeng, Hane Lee, Ascia Eskin, Stanley F Nelson, Formal analysis, Investigation; William H Xie, Data curation, Supervision; Samah Al-Tawalbeh, Formal analysis, Validation, Investigation; Mohammad El-Khateeb, Mohammad Shboul, Bruno Reversade, Resources; Mahmoud A Pouladi, Resources, Investigation; Mohammed Al-Raqad, Resources, Funding acquisition

### Author ORCIDs

Poh Hui Chia  http://orcid.org/0000-0001-8070-112X
Franklin Lei Zhong  http://orcid.org/0000-0002-0516-6021
Mahmoud A Pouladi  https://orcid.org/0000-0002-9030-0976
Bruno Reversade  http://orcid.org/0000-0002-4070-7997

### Ethics

Human subjects: Informed consent was obtained from the families for genetic testing in accordance with approved institutional ethical guidelines (Institute of Medical Biology, Singapore, A*STAR, Singapore, NUS-IRB reference code 10-051). Parent of the patients has signed the eLife Consent to Publish Form and is available if necessary.

### Decision letter and Author response

Decision letter https://doi.org/10.7554/eLife.32451.017
Author response https://doi.org/10.7554/eLife.32451.018

## Additional files

### Supplementary files

• Supplemental file 1. Statistical Data Analysis
DOI: https://doi.org/10.7554/eLife.32451.012

• Source code 1. IBD linkage program.

DOI: https://doi.org/10.7554/eLife.32451.013

• Transparent reporting form
DOI: https://doi.org/10.7554/eLife.32451.014

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
