## [Decision Letter]

Thank you for submitting your article "A homozygous loss-of-function *CAMK2A* mutation causes growth delay, frequent seizures and severe intellectual disability" for consideration by *eLife*. Your article has been reviewed by three peer reviewers, and the evaluation has been overseen by a Reviewing Editor and K VijayRaghavan as the Senior Editor. The following individuals involved in review of your submission have agreed to reveal their identity: Karl Giese (Reviewer #1); A.J Robison (Reviewer #3).

The reviewers have discussed the reviews with one another and the Reviewing Editor has drafted this decision to help you prepare a revised submission.

All reviewers have substantive comments, most of which are non-overlapping. Some of the comments require you to carry out new experiments. For submitting a revised version, it is essential that you address all the comments of all the reviewers. Along with your revised submission, please attach a point-by-point statement of the actions taken by you. If you disagree with any of the points raised by the reviewers, please state so with justification in your actions-taken-report.

Reviewer #1:

This manuscript is of fundamental importance for the CaMKII field, as for the first time it makes a convincing case that CaMKII dysfunction is relevant for brain diseases. The strength of the manuscript is that a family is described where two children are homozygous for a missense mutation in CaMKIIalpha. Biochemical characterisation of the missense mutation strongly suggests that this mutation causes loss-of function, by not only preventing oligomerisation of CaMKIIalpha subunits, but also mislocalising CaMKIIalpha to the nucleus and enhancing the proteasomal dedragation of the kinase subunits. I think that this work is important that it deserves to be published in *eLife*. I have some suggestions that can further strengthen the manuscript:

1) I could not find any data regarding impaired intellectual disability. If possible, could such data be included?

2) Regarding the table in Figure 1—figure supplement 1, it would be amazing if data for heterozygous and wild-type children could be included. This would provide very strong evidence that specifically the missense mutation causes the observed phenotypes. Would you have patient consent to include such information?

3) The authors found that the identified missense mutation mislocalizes CaMKIIalpha to nucleus in a non-neuronal cell line. If possible, it would be good to show that this also happens in primary neurons. This would suggest that CaMKIIalpha with the missense mutation is not delivered to synapses, substantially blocking synaptic plasticity.

4) Unfortunately, the presented in vivo studies with *C. elegans* focus on the wrong neuronal cell type. In mammals CaMKIIalpha is expressed exclusively in glutamatergic neurons (not cholinergic motoneurons). Therefore, I think that Figure 2 must be removed from the manuscript, as the findings are misleading. -- In my opinion, for this manuscript it is not necessary to include in vivo data. However, if authors wish to include in vivo data, then they should focus on the correct neuronal cell type in any in vivo system.

Reviewer #2:

Chia and colleagues have authored a potentially important manuscript identifying an association between a novel homozygous missense mutation in CAMK2A and global neurodevelopmental delay with severe intellectual disability, seizures, and delayed growth. Although interesting, a number of major issues need to be addressed:

1) Substantially more details are required regarding the genetic analysis, in order to better establish the likely causality of the candidate CAMK2A mutation. In particular, the workflow by which the proposed causative variant has been identified needs to be much better clarified.

a) Can the authors provide additional pedigree data regarding the parental consanguineous relationship? If not, then on what basis are the parents reported as consanguineous?

b) The authors should provide the genome-wide results of the homozygosity mapping prior to filtering out homozygous segments found in the unaffected family members, and prior to filtering for regions of homozygosity >5cM.

c) On what basis have the authors established the "confidence criteria to identify IBD blocks were a minimum of 5 cM"?

d) Regarding the variants identified by exome sequencing, the authors should minimally provide a list of all 72 homozygous protein-coding variants with MAF<1%, and sufficient information to determine their localization within the regions of homozygosity in all of the genotyped family members.

2) Functional experiments: a) The authors convincingly establish that the H477Y mutation blocks oligomerization of the CAMK2A holoenzyme. However, they subsequently attempt to evaluate protein stability using MG132. I have some difficulties with this experiment.

Specifically, how can it be that in the DMSO treated lysates (Figure 2—figure supplement 2C), such big differences are found between the different CAMK2A construct expression levels, whereas in Figure 1H the expression levels are similar?

Was each experiment done at different time points after transfection?

How do the authors control for transfection efficiency? Based on the results shown here, in the absence of a demonstration that the transfection efficiency was equal across the various constructs, it is difficult to reach a firm conclusion regarding the stability or expression levels of CAMK2A [H477Y].

b) Figure 2C and 2E: The images provided are consistent with a rescue of the synaptic phenotype when expressing either Unc-43 or wild-type human CAMK2, but not with the H477Y mutant. However, the experimental design is unclear.

The legend reports n>80 (Figure 2C) and n=100 (Figure 2E). But what does 'n' represent (e.g., is this the number of puncta, images, or worms measured)? For Figure 2C, why is it not possible to provide an exact value?

c) Figure 2C: Why do the authors use a Student's t-test for these data? If they want to compare the different conditions, a one-way ANOVA with post-hoc correction should be applied.

d) Figure 2D: By what criteria have the authors defined the asynaptic region?

e) The Unc-43 worm has clear phenotypes, as the authors themselves describe in the Results section: "Null mutants for unc-43 are flaccid in posture and move with a flattened uncoordinated waveform." These phenotypes are very clear. Why didn't they try rescuing these phenotypes? This would have provided a much stronger opportunity to assess the functional relevance of the rescue experiments.

f) What is the relevance of examining the localization of the association domain? The authors reason that since the mutation blocks oligomerization, it is more likely to diffuse through the nuclear pore, thereby reducing cytosolic accumulation. But does the association domain itself oligomerize in HEK293T cells?

The authors seem to infer on the basis of these experiments that CAMK2A is likely to be mislocalized in the homozygous H477Y carriers. However, they would need to show the same experiment using full-length reference and mutant CAMK2A in order to establish more definitive evidence.

Reviewer #3:

The manuscript "A homozygous loss-of-function CaAMK2A mutation causes loss of growth, frequent seizures, and severe intellectual disability" uses sequencing of patient DNA, biochemistry, and *C. elegans* phenotyping to characterize a novel coding mutation preventing holoenzyme formation in CaMKIIalpha protein in humans. The authors demonstrate that an autosomal recessive mutation in a histidine residue critical for association domain interactions is associated with disease state in a human family, then go on to use traditional biochemistry to show that this mutant kinase forms interactions with WT but not with itself, a finding beautifully consistent with the recessive nature of the disease state. Further, they use a rescue strategy in a *C. elegans* model lacking expression of the CaMKII homolog to show that neither the human nor the worm kinase with this mutation can rescue worm phenotype, while either WT kinase does so. Because CaMKII has long been associated with cognition through mostly mouse studies but has never before been directly and causally tied to a human cognitive disorder, this is a very exciting finding that will help to validate and reinvigorate study of this essential molecule. The paper is clear and well-written, the experiments directly address the hypotheses and are appropriately interpreted (for the most part – see below), and the findings are certain to strongly impact the field. I have some minor concerns below about interpretation and mechanism, and one somewhat larger concern:

The *C. elegans* experiments are very convincing. However, I wonder whether the mutant and WT CaMKII proteins used in the rescue experiments have the same localization. This ought to be very easy to determine by immunofluorescence, even if the constructs aren't tagged. Does the mutant accumulate in the nucleus of the *C. elegans* neurons as it does in HEK cells? Does the WT adopt a postsynaptic localization? Data such as these would greatly strengthen the manuscript.

Overall, I am highly enthusiastic about this manuscript.

---

## [Author Response]

Reviewer #1:This manuscript is of fundamental importance for the CaMKII field, as for the first time it makes a convincing case that CaMKII dysfunction is relevant for brain diseases. The strength of the manuscript is that a family is described where two children are homozygous for a missense mutation in CaMKIIalpha. Biochemical characerisation of the missense mutation strongly suggests that this mutation causes loss-of function, by not only preventing oligomerisation of CaMKIIalpha subunits, but also mislocalising CaMKIIalpha to the nucleus and enhancing the proteasomal degradation of the kinase subunits. I think that this work is important that it deserves to be published in eLife. I have some suggestions that can further strengthen the manuscript:1) I could not find any data regarding impaired intellectual disability. If possible, could such data be included?

The clinicians cannot obtain a definitive IQ test since both affected siblings are non-verbal and do not respond to verbal cues. We believe that the supplementary videos do convey the extent and severity of the ID in both affected siblings.

2) Regarding the table in Figure 1—figure supplement 1, it would be amazing if data for heterozygous and wild-type children could be included. This would provide very strong evidence that specifically the missense mutation causes the observed phenotypes. Would you have patient consent to include such information?

We have confirmed that the healthy siblings (who are either heterozygous or non-carrier for the CAMK2A ^H477Y^ allele) do not display any signs of neuro-developmental delay. We have updated the clinical table in Figure 1—figure supplement 1 to include all five siblings.

3) The authors found that the identified missense mutation mislocalizes CaMKIIalpha to nucleus in a non-neuronal cell line. If possible, it would be good to show that this also happens in primary neurons. This would suggest that CaMKIIalpha with the missense mutation is not delivered to synapses, substantially blocking synaptic plasticity.

We have significantly revised our initial biochemical analyses of overexpressed CAMK2A mutant. In the initial submission, the nuclear localization was shown using GFP-tagged to the CAMK2A association domain, but not with the full length CAMK2A protein. We did not mean to claim that full length CAMK2A would mislocalize in vivo. Instead, we simply used nuclear exclusion as an in vitro assay to test the oligomerization of CAMK2A association domain.

We have removed these data to avoid any confusion and replaced them with a more rigorous functional experiment using neurons derived from the proband’s iPS cells. These new results demonstrate a profound impairment in neuronal activity in patient’s neurons that are cultured in vitro (Figure 2) suggesting that endogenous CAMK2A^H477Y^ is most likely dysfunctional.

We have attempted to decipher the localization of endogenous CAMK2A by immunofluorescence in iPSC-derived neurons. However, all five commercial CAMK2A antibodies tested failed to give a single band by Western blot using lysates from wild-type neurons (Author response image 1). Likewise, these antibodies could not reflect the synaptic localization of wild-type CAMK2A in iPSC-derived neurons using IF. We therefore could not carry out a rigorous evaluation of the localization of the CAMK2A^H477Y^ in iPSCs-derived neurons.

4) Unfortunately, the presented in vivo studies with C. elegans focus on the wrong neuronal cell type. In mammals CaMKIIalpha is expressed exclusively in glutamatergic neurons (not cholinergic motoneurons). Therefore, I think that Figure 2 must be removed from the manuscript, as the findings are misleading. -- In my opinion, for this manuscript it is not necessary to include in vivo data. However, if authors wish to include in vivo data, then they should focus on the correct neuronal cell type in any in vivo system.

We understand your concern that in mammals, CAMK2A is mostly found in glutamatergic neurons. In vertebrates, CAMKII isoforms are encoded by at least four genes (⍺, β, γ, δ). In the worm, the *unc-43* gene encodes only one CaMKII orthologue which is found in all neurons. Mutations in *unc-43* cause multiple behavioral defects in locomotory activity, in the clock control of defecation, in the regulation of body-wall muscle excitation and spontaneous activity (David et al., 1999). Thus in *C. elegans, unc-43* can perform all these functions regardless of the neuronal cell type in which it is expressed.

CAMK2 is highly conserved from humans to worm. We show in Figure 4C and 4F that the human CAMK2A orthologue is able to fully rescue synaptic defects due to the loss of *unc-43* in the DA9 neuron. This indicates that the human CAMK2A functions similarly in cholinergic neurons of *C. elegans*. More importantly, the identified patient mutation p.H477Y when introduced in either UNC-43 or CAMK2A, is unable to rescue the *unc-43* synaptic defects.

To further strengthen this data, we have performed behavioural rescue experiments by expressing either wildtype or H466Y mutated UNC-43 protein in worms defective for *unc-43.* We have used a pan-neuronal promoter to express UNC-43 in all neurons and scored the phenotype in a double-blind experimental setup. Consistently, we observe that the wildtype UNC-43, but not UNC-43^H466Y^, is able to fully rescue all behavioral defects of *unc-43* worms. This is providing further evidence that the patient identified mutation renders CAMK2A non-functional in multiple in vivo assays.

We believe the *C. elegans* data are key to prove the pathogenicity of this human allele. They are now complemented by functional assays using patient IPSCs-derived neurons (Figure 2).

Reviewer #2:Chia and colleagues have authored a potentially important manuscript identifying an association between a novel homozygous missense mutation in CAMK2A and global neurodevelopmental delay with severe intellectual disability, seizures, and delayed growth. Although interesting, a number of major issues need to be addressed:1) Substantially more details are required regarding the genetic analysis, in order to better establish the likely causality of the candidate CAMK2A mutation. In particular, the workflow by which the proposed causative variant has been identified needs to be much better clarified.

We have included in this revised manuscript, a more detailed description of the pipeline, including IBD mapping, whole-exome sequencing and data filtering. We are adhering to our standard operating procedures which have allowed us to genetically diagnose numerous Mendelian disorders over the years (Reversade et al., 2009; Bonnard et al., 2012; Hu and Pomp, 2014; Cetinkaya and Xiong, 2016; Zhong et al., 2016; Gordon and Xue, 2017).

a) Can the authors provide additional pedigree data regarding the parental consanguineous relationship? If not, then on what basis are the parents reported as consanguineous?

We could not obtain additional medical record from members of the extended family. We confirm that parents are first cousins as stated by them during several clinical visits. This is independently verified from the SNP genotyping analysis of their 5 children. On average, 10% of the children’s genomes are homozygous- proving that their parents are at least first cousins.

% of homozygosity in all children from familyChildMb% homozygousII-1309.410.3II-2319.210.6II-3378.212.6II-4310.210.3II-5183.16.1

b) The authors should provide the genome-wide results of the homozygosity mapping prior to filtering out homozygous segments found in the unaffected family members, and prior to filtering for regions of homozygosity >5cM.

These data are now provided in Supplemental file 1 and Figure 1—figure supplement 1D.

c) On what basis have the authors established the "confidence criteria to identify IBD blocks were a minimum of 5 cM"?

IBD regions <5 cM are likely to be false positives. According to Duran et al., (2014), "most 2–3 cM segments are erroneous and only segments longer than 5 cM have a negligible number of false positives".

d) Regarding the variants identified by exome sequencing, the authors should minimally provide a list of all 72 homozygous protein-coding variants with MAF<1%, and sufficient information to determine their localization within the regions of homozygosity in all of the genotyped family members.

These data are now provided in Figure 1—figure supplement 1E, with a thorough description in the text. Briefly, four homozygous variants from the proband’s exome were identified within the chromosome 5 IBD candidate block. Notably, homozygotes for these variants, except the CAMK2A^H477Y^ allele, have been identified in public sequencing databases, such as ExAC and GnomAD, and therefore can be ruled out.

2) Functional experiments: a) The authors convincingly establish that the H477Y mutation blocks oligomerization of the CAMK2A holoenzyme. However, they subsequently attempt to evaluate protein stability using MG132. I have some difficulties with this experiment.Specifically, how can it be that in the DMSO treated lysates (Figure S2C), such big differences are found between the different CAMK2A construct expression levels, whereas in Figure 1H the expression levels are similar?Was each experiment done at different time points after transfection?How do the authors control for transfection efficiency? Based on the results shown here, in the absence of a demonstration that the transfection efficiency was equal across the various constructs, it is difficult to reach a firm conclusion regarding the stability or expression levels of CAMK2A [H477Y].

Thank you for encouraging us to improve these results. These experiments have been significantly revised with improved constructs consisting of a single cistronic T2A-mCherry cassette to control for both transfection and translation efficiency. These results are described in Figure 3E, 3F and Figure 2—figure supplement 2B.

b) Figure 2C and 2E: The images provided are consistent with a rescue of the synaptic phenotype when expressing either Unc-43 or wild-type human CAMK2, but not with the H477Y mutant. However, the experimental design is unclear.The legend reports n>80 (Figure 2C) and n=100 (Figure 2E). But what does 'n' represent (e.g., is this the number of puncta, images, or worms measured)? For Figure 2C, why is it not possible to provide an exact value?

We have made changes in the legend text to clarify the exact parameter of measurement for each experiment. For instance, “error bars represent SEM with number of synaptic puncta quantified n> 80” and “Graph shows the percentage of animals with the WT and *unc-43* mutant phenotypes with n=100 animals scored for each line”.

c) Figure 2C: Why do the authors use a Student's t-test for these data? If they want to compare the different conditions, a one-way ANOVA with post-hoc correction should be applied.

The statistics for experiments comparing multiple conditions has been revised according to your suggestions. We have implemented a one-way ANOVA test with either Bonferroni post-hoc test or two-way ANOVA with Tukey post-hoc test.

d) Figure 2D: By what criteria have the authors defined the asynaptic region?

The asynaptic region is the defined as the part of the axon extending out from the DA9 cell body to the first *en passant* synapse located on the dorsal part of the axon. It has been shown that the positional information required to define this asynaptic region is controlled by *lin-44*/Wnt secreted by four hypodermal cells in the tail of the worm (Klassen and Shen, (2007). This asynaptic region of the axon is also defined by different synaptic vesicle transport dynamics as compared to the dendrite and distal axon of DA9 (Maeder et al., (2014).

e) The Unc-43 worm has clear phenotypes, as the authors themselves describe in the Results section: "Null mutants for unc-43 are flaccid in posture and move with a flattened uncoordinated waveform." These phenotypes are very clear. Why didn't they try rescuing these phenotypes? This would have provided a much stronger opportunity to assess the functional relevance of the rescue experiments.

We have carried out a behavioral rescue as shown Figure 3G: in contrast to UNC-43^wild-type^, UNC-43^H466Y^ which is homologous to CAMK2A^H477Y^, completely failed to rescue the uncoordinated and convulsive phenotype of the mutant *unc-43* worms. Thank you for suggesting this crucial experiment which provides further support to our initial observation.

f) What is the relevance of examining the localization of the association domain? The authors reason that since the mutation blocks oligomerization, it is more likely to diffuse through the nuclear pore, thereby reducing cytosolic accumulation. But does the association domain itself oligomerize in HEK293T cells?

We have removed these data to avoid any confusion. Our biochemical data using native PAGE and co-immunoprecipitation (Figure 3C,D) support the oligomerization defect when CAMK2A was expressed in 293T cells as well as when translated in vitro.

The authors seem to infer on the basis of these experiments that CAMK2A is likely to be mislocalized in the homozygous H477Y carriers. However, they would need to show the same experiment using full-length reference and mutant CAMK2A in order to establish more definitive evidence.

These experiments are now shown in Figure 3E and Figure 2—figure supplement 2B, with full length CAMK2A. We focused on the most striking difference, which was the reduction in overall level in mutant CAMK2A.